# Recent Applications of Melanin-like Nanoparticles as Antioxidant Agents

**DOI:** 10.3390/antiox12040863

**Published:** 2023-04-02

**Authors:** Alexandra Mavridi-Printezi, Arianna Menichetti, Dario Mordini, Riccardo Amorati, Marco Montalti

**Affiliations:** 1Department of Chemistry «Giacomo Ciamician», University of Bologna, Via Selmi 2, 40126 Bologna, Italydario.mordini2@unibo.it (D.M.);; 2Tecnopolo di Rimini, Via Dario Campana 71, 47921 Rimini, Italy

**Keywords:** melanin, nanoparticles, polydopamine, ROS, allomelanin, fluorescence, nanomedicine, radicals, bioimaging, ferroptosis

## Abstract

Nanosized antioxidants are highly advantageous in terms of versatility and pharmacokinetics, with respect to conventional molecular ones. Melanin-like materials, artificial species inspired by natural melanin, combine recognized antioxidant (AOX) activity with a unique versatility of preparation and modification. Due to this versatility and documented biocompatibility, artificial melanin has been incorporated into a variety of nanoparticles (NP) in order to give new platforms for nanomedicine with enhanced AOX activity. In this review article, we first discuss the chemical mechanisms behind the AOX activity of materials in the context of the inhibition of the radical chain reaction responsible for the peroxidation of biomolecules. We also focus briefly on the AOX properties of melanin-like NP, considering the effect of parameters such as size, preparation methods and surface functionalization on them. Then, we consider the most recent and relevant applications of AOX melanin-like NPs that are able to counteract ferroptosis and be involved in the treatment of important diseases that affect, e.g., the cardiovascular and nervous systems, as well as the kidneys, liver and articulations. A specific section will be dedicated to cancer treatment, since the role of melanin in this context is still very debated. Finally, we propose future strategies in AOX development for a better chemical understanding of melanin-like materials. In particular, the composition and structure of these materials are still debated, and they present a high level of variability. Thus, a better understanding of the mechanism behind the interaction of melanin-like nanostructures with different radicals and highly reactive species would be highly advantageous for the design of more effective and specific AOX nano-agents.

## 1. Introduction

Atmospheric molecular oxygen is the basis of life on Earth, and its generation in the Great Oxidation Event (GOE) is considered to be also the actual origin of the formation of the first cellular living species [1,2]. The combination of molecular oxygen with organic molecules, known as oxidation, is essential for the exploitation of the chemical energetic content of organic molecules that are acquired by living organisms as nutrients. Even if oxidation is a process necessary for life, oxygen consumption is also related to an organism’s aging, diseases, and death [3,4,5,6]. The main factor causing these events is that the reduction of oxygen, that if completed would lead to the production of totally harmless water molecules, is a multi-electron process (as schematized in Figure 1) and involves the formation of reactive, and potentially aggressive, intermediates: the Reactive Oxygen Species (ROS). Besides, not all ROS are radicals, but they are the main party responsible for the formation of radicals in living organisms, including humans [7]. Free radicals can play a disruptive function since they are responsible for the oxidative damage of biomolecules, thus causing the degradation of proteins, lipids, or even nucleic acids [8]. Nevertheless, radicals and related species are necessary regulatory agents in essential biological processes, including gene expression, cell proliferation, apoptosis, phosphorylation or calcium concentration control, cell division, and the elimination of microorganisms [9,10,11]. Consequently, the total removal of radicals from living cells and organisms would be as dangerous for their health as their over-production. Therefore, optimal organism functionality requires the achievement of a perfect redox balance via the proper regulation of the process of radical formation and degradation; this is known as “redox homeostasis”. When redox balance is achieved, the organism is found in a state in which finely regulated systems maintain very low controlled levels of ROS [7,9]. Deviation from this balance and an uncontrolled increase in the radical content is known as “Oxidative Stress”, and it is associated with severe pathologies such as cancer [12], SARS-CoV-2 infection [13], diabetes mellitus [14], skin injuries [15,16,17], and kidney [18], liver [19] cardiovascular [20] and neurodegenerative diseases [21,22].

### 1.1. Generation of Radicals in Living Organisms

Under aerobic conditions, living organisms reduce more than 90% of oxygen consumed directly to water without ROS generation. The four-electron reduction process (see Figure 1) is carried out by cytochrome oxidase in an electron transport chain (ETC) and it is coupled with oxidative phosphorylation to produce energy in the form of ATP. Indeed, the full oxygen reduction process fails, culminating in the one-electron reduction of O_2_; this gives rise to the formation of the superoxide anion radical O_2_^•−^, but only for the 10% of the totally consumed oxygen. Further, one-electron reduction and double protonation yield hydrogen peroxide (H_2_O_2_) that, importantly, is not a radical, but is a stronger oxidant than O_2_ and can generate a very reactive hydroxyl radical (HO^•^) via further reduction with the concomitant formation of the hydroxyl anion (HO^−^). The main reaction of HO^•^ in biological systems is the hydrogen atom transfer (HAT) or radical addition (RA) to double bonds at the expense of different compounds such as proteins and lipids. It is very important to underline that this HAT or RA first event not only leads to the formation of other radicals such as R^•^, RO^•^, ROO^•^ (where R is an organic residue), but it can initiate a chain reaction, known as propagation, that amplifies the damage and leads to the gradual oxidation of other molecules, as schematized in Figure 1 [23].

Propagation of radical chain reactions in living organisms can produce dramatic effects, such as as lipid peroxidation and cell death in ferroptosis [24]. It is worth noting that the radical propagation involves the consumption of molecular oxygen, and it is only mediated by radicals. This is consistent with the thermodynamic instability of hydrocarbons in oxygenated environments. Because of this, any organic matter, including living organisms, would “burn” spontaneously in the presence of oxygen, while their existence is only kinetically guaranteed. Hence, it should be remembered that radicals, as catalyzers, unleash the oxidative action of oxygen by providing a kinetically accessible path for the degradation [7].

Regarding ROS, over 90% of these species are produced by mitochondria in eukaryotic cells, mostly due the escape of electrons from mitochondrial ETC to molecular oxygen, resulting in O_2_^•−^ that is converted to H_2_O_2_ or HO^•^. The amount of ROS produced in this stage is not controlled by the cells, and in order to keep the level of ROS low, a finely regulated antioxidant system is needed, as discussed in Section 1.2. Additionally, minor amounts of ROS are produced, with a similar mechanism, by ETC located in/at the endoplasmic reticulum, plasmatic, and nuclear membranes, as well as by some oxidases. Finally, one more ROS source is related to the autoxidation of different small molecules of endo- and exogenous origin.

In addition to ROS, other important reactive species that are generated in living organisms include Reactive Nitrogen Species (RNS) [25]. In particular, NO^•^ is produced enzymatically by nitric oxide synthases (NOS) that convert L-arginine into L-citrulline and NO^•^, via a 5-electron oxidation of a guanidine nitrogen of L-arginine. In addition to its toxicity, NO^•^ plays a role in the controlled concentration, important signaling and regulation functions of cells. NO^•^ has a half-life of only a few seconds, but it possesses a greater stability at lower oxygen concentrations. The simultaneous formation of both the superoxide anion and nitric oxide may produce significant amounts of a much more oxidatively active molecule, the peroxynitrite anion (ONOO^−^), which is a potent oxidizing agent that can cause DNA fragmentation and lipid oxidation [7].

### 1.2. Regulation of Radicals in Living Organisms

Living organisms exploit multilevel and sophisticated systems in order to defend against the excessive production of radicals. Molecules that are able to reduce the concentration of radicals are called antioxidants (AOXs) and can be classified as preventive and chain-breaking AOXs, depending on their mechanism of action in the autoxidation radical chain [23]. Other classifications might also be used, such as enzymatic and non-enzymatic, endogenous and exogenous [20], or “primary” and “secondary”, depending on whether they react directly with ROS, or if they support other endogenous AOXs [26,27] (Table 1).

#### 1.2.1. Preventive Antioxidants

Molecules, enzymes, and materials that prevent the initial formation of radicals can be regarded as preventive AOXs. The Fenton reaction, consisting of the one-electron reduction of hydroperoxides (H_2_O_2_ or ROOH) to RO^•^ or HO^•^ radicals by reduced metal ions such as Fe^2+^ and Cu^+^, is one of the most important sources of radicals in the cellular milieu. As a consequence, all the systems able to remove “free” metal ions (i.e., metal chelators) and hydroperoxides have an important AOX effect [36]. H_2_O_2_ can be decomposed to water and oxygen by catalase (CAT) or can be reduced to water by glutathione peroxidase enzymes (GPX). CAT is a tetrameric ferriheme oxidoreductase that is mainly located in peroxisomes and that is most active in the liver and red blood cells. GPX is a family of selenium-dependent oxidoreductases, which use H_2_O_2_ or organic hydroperoxide as the oxidant, and the tripeptide glutathione GSH as the electron donor. GPX4, specifically, is a phospholipid hydroperoxidase located in the cellular membrane that protects cells from membrane lipid peroxidation and from ferroptosis cell death [37]. At the basis of the antioxidant activity of GPX, there are molecules such as reduced NADPH, which are produced by other enzymes that are not directly involved in the antioxidant action and therefore act as co-enzymes.

Radical chains could potentially be initiated by the reaction of the hydroperoxyl radical (HOO^•^), which at physiological pH, is present in small amounts due to the protonation of superoxide (O_2_^•−^) (pK_a_(HOO^•^) = 4.7) [38]. However, the concentration of O_2_^•−^ in the cell is strictly controlled by the activity of superoxide dismutase (SOD), which catalyzes the dismutation of O_2_^•−^ into molecular oxygen (O_2_) and hydrogen peroxide (H_2_O_2_). SODs can be divided into four groups, depending on the metal cofactors. Copper-zinc SOD is most abundant in chloroplasts, cytosol, and extracellular space. Iron SOD is found in plant cytosol and in microbial cells, whereas manganese SODs are mitochondrial. By consuming O_2_^•−^, SOD also indirectly reduces the formation of peroxynitrite (ONOO^−^, reaction 2), and increases the biological availability of NO^•^.

#### 1.2.2. Radical-Trapping Antioxidants (RTA)

Molecules able to slow down autoxidation by trapping the propagating radicals are named “chain-breaking”, or radical-trapping AOXs (RTAs). In the biological environment, they are mainly represented by small molecules such as ascorbate, α-tocopherol and ubiquinol (reduced Coenzyme Q) [23]. The mechanism of action of this class of AOXs is to trap alkylperoxyl radicals (ROO^•^ in Figure 1) faster than ROO^•^ can react with the substrate. Moreover, after reacting with ROO^•^, RTAs form stable radicals that do not propagate the oxidative chain [23]. It should be noted that the trapping of R^•^, RO^•^ or HO^•^ radicals is not relevant for describing the RTA activity, because these radicals have an extremely short lifetime under most physiological conditions; thus, only the capability to trap ROO^•^ has to be considered [36]. This is particularly important when chemical methods are employed for the evaluation of the activity of a RTAs, as the limitations and shortcomings of many simplified assays to predict their action have to be considered [39,40].

Radical-trapping AOXs usually act in a synergistic fashion, in which a lipid-soluble RTA, able to stop chain propagation in a lipid environment, is regenerated by hydrosoluble RTAs or by specific enzymatic systems. Interestingly, there is growing evidence that superoxide, being a two-faced oxidizing and reducing radical, can act synergistically with RTAs, thus extending their duration time and providing reducing equivalents for the catalytic removal of ROO^•^ radicals [41]. Some RTA, such as nitroxides and ortho-quinones, act as SOD-mimics by being cyclically reduced by O^•−^/HOO^•^ (depending on the pH) and oxidized by ROO^•^ or OO^•−^/HOO^•^ radicals [42,43,44].

Some nanosized materials present unique properties [45,46,47,48] including preventive or radical trapping activity possess characteristics that are similar to the aforementioned enzymes and are, therefore, called nanozymes [49]. These nano-AOXs present several advantages with respect to other molecular AOXs [50]. Their main advantages are, evidently, their versatility and modulability, since the intrinsic AOX properties of any nanomaterial highly depends on the particle size, the surface charge, the porosity, and the surface coating. Other advantages that plenty of nanomaterials exhibit include increased bioavailability, controlled release, and targeted delivery to the site of action.

Melanin-like nanomaterials present good AOX properties combined with a high and documented biocompatibility. In this review, we will discuss the most recent developments in the application of melanin-like nanoparticles (NP) for the AOX treatment of oxidative stress-related pathologies. We will also focus briefly on the AOX properties of melanin-like NPs, considering the effect of parameters such as size, the preparation method and surface functionalization on their activity. The objective of this review paper is to show that melanin, melanin-like and melanin-containing NPs can be efficiently used for AOX therapy in several serious diseases. This study was designed by focusing on the most recent examples of relevant scientific papers reporting on new nanoplatforms based on melanin-like materials, and in particular NPs, for AOX therapy. Particular attention was dedicated to scientific works published in the last five years.

Discussing the applications of melanin-based nanomaterials for the treatment of different pathologies, the methods used to determine and evaluate their AOX activity will be also reported [36,51,52,53]. A specific section will be dedicated to cancer treatment since the role of melanin in this context is still highly debated.

## 2. AOX Properties of Melanin NP

Melanin is a family of polymeric pigments that is present in organisms, where they perform different functions including pigmentation, radical scavenging, radiation protection, and thermal regulation [54]. These materials, depending on the various chemical precursors used in their biosynthesis, are classified into eumelanin, pheomelanin, neuromelanin, allomelanin, and pyomelanin [55,56,57,58,59,60,61,62,63,64,65]. This classification of melanin is mostly based on the kind of molecular precursor involved in their bio-synthesis and was recently reviewed by Gianneschi’s group [54]. It is very interesting to underline that melanin is one of the few materials that is already present in organisms in the form of nanoaggregates. Although the AOX properties of natural melanin are well documented [66,67,68,69,70,71,72,73,74,75,76,77], and in addition to the ubiquitous presence of melanin in nature, its most recent and promising application is based on the development of synthetic analogs [78]. The main reason for this is that melanin-like materials can be easily produced from low cost and biocompatible precursors via very versatile and environmentally friendly processes [79,80]. Importantly, melanin synthetic analogues maintain the intrinsic biocompatibility of the natural pigments. The most exploited and popular form of artificial melanin, also called melanin-like materials, is surely polydopamine (PDA), resulting from the oxidation and polymerization of dopamine upon simple exposure to atmospheric oxygen in an alkaline aqueous environment. PDA is typically produced in the form of NPs and it has been reported to present good AOX properties [55,81,82]. We would like to stress that due to their high biocompatibility and dispersibility in physiological media, melanin-like NPs can be typically directly administered to organisms for bio-application purposes without requiring any delivery platform, being also themselves good vectors for the delivery of other bioactive agents [79,80].

The mechanism behind the AOX activity of PDA NPs has been proposed by Amorati and colleagues, who showed that these artificial melanin NPs become an excellent trap for ROO^•^ (alkylperoxyl radicals), which are radicals typically formed during the autoxidation of lipid substrates in the presence of hydroperoxyl radicals (HOO^•^). As schematized in Figure 2, during the reaction, the orthoquinone moiety of PDA is reduced by the reaction with HOO^•^ via a H-atom transfer mechanism [83]. The same author demonstrated that the AOX spectrum of PDA can be further expanded by conjugation to nitroxide [84].

Ferroptosis is a recently discovered cell death pathway involved in degenerative diseases, as well as in tumor cells suppression. Cell death is due to uncontrolled lipid peroxidation of cell membranes resulting from the simultaneous presence of free Fe^2+^ ions and lipid hydroperoxides [37]. PDA NP (diameter ≈ 220 nm) have been proven to be able to protect the heart against ischemia/reperfusion injury by inhibiting ferroptosis, thanks to the Fe^2+^ chelating and radical trapping activity of PDA [85].

The effect of the NP size on the AOX activity of PDA was investigated by Carmignani et al. [86], who synthesized a set of differently sized NPs (from 150 to 960 nm) by controlling the ammonia/dopamine molar ratio during the preparation procedure. Electron paramagnetic resonance (EPR) spectroscopy coupled with the spin-trapping technique was used to analyze the effect of the size on the AOX activity of the melanin-mimic NPs. The Fenton reaction (Fe^2+^ + H_2_O_2_ → Fe^3+^ + OH^•^ + HO^−^) was started in situ for each sample, and free hydroxyl radicals present in the sample were trapped using DMPO as a spin trap. For each of the differently sized classes of NP, the EPR spectrum was obtained after 5 min from the start of the Fenton reaction. Quantitative results showed an OH^•^ radical scavenging efficiency of 92.2% for the smallest NP; this value decreased dramatically by increasing the size, becoming less than 10% for the largest PDA NP.

Ball and coworkers investigated the effect of the oxidant used for the preparation of PDA films on their AOX properties [87]. These authors exploited a combination of experimental techniques, such as as atomic force microscopy, cyclic voltammetry, and X ray photoelectron spectroscopy, in order to evaluate the effect of the structural features and chemical composition on the AOX properties of the films, which was proven via the DPPH discoloration method. DPPH is a purple-colored radical species that upon reduction, in the presence of an oxidant, undergoes a color change from purple to yellow, which can be easily followed by spectrophotometry. The conclusion was that the AOX properties of PDA films are not only dependent on the type of the employed oxidant, which can be expected to affect both the density of the oxidizable groups on the surface of the PDA and the oxidant film’s morphology and roughness. The authors also proposed models for the mechanism of dopamine oxidation during the formation of PDA. PDA is also very efficient in complexing several metal ions and the effect of the presence of metals on the AOX properties has been recently reviewed [88]. Similarly, the general effect of surface functionalization on the ROS-scavenging ability has been discussed [89].

The AOX activity of Mesoporous PDA was also recently reported, testing the DPPH radical scavenging activity [90]. The main advantage of these porous materials is that they can be efficiently loaded with therapeutic molecular agents.

PDA is not the only kind of melanin-related material that presents AOX activity. Indeed, the AOX properties of fungi-derived melanin were also demonstrated [91,92,93]. Since fungi melanin is based on the precursor 1,8-dihydroxynaphthalene, synthetic analogs of this natural type of melanin, called allomelanin, were also prepared, demonstrating good AOX properties [60,94]. In the case of neuromelanin, its role in oxidative stress is complicated. In fact, humans’ catecholaminergic neurons gradually become black during ageing because of the accumulation of neuromelanin, which serves as a very efficient quencher for toxic molecules. However, when a neuron degenerates, neuromelanin is released and it contributes to exacerbating the oxidative stress that ultimately leads to the neurodegenerative process [95]. The AOX activity of a synthetic analog of pyomelanin was also demonstrated [96]. According to the authors, soluble pyomelanin mimics can represent a suitable alternative to the insoluble melanin pigment for biotechnological applications.

## 3. Protection from Radiation Induced ROS

Ultraviolet (UV) exposure is known to lead to the generation of ROS, resulting in cell death [97,98]. Together with DNA lesions, like cyclobutane-pyrimidine dimers (CPDs), 6-4 photoproducts (6-4PPs) and their Dewar valence isomers [99], light-induced ROS generation is a major issue when human skin is exposed to solar radiation; this can be partially reduced by naturally extracted melanin [100]. Gianneschi and coworkers showed that synthetic PDA NPs can be used to prevent UV-induced cellular damage in human epidermal keratinocytes (HEKa) [101]. In particular, as schematized in Figure 3, NPs synthesized by the simple polymerization of dopamine an in alkaline environment are up-taken by HEKa cells that mimic the behavior of natural melanosomes in terms of cellular distribution and also prevent photoinduced ROS generation. The ROS that were produced in response to UV irradiation were detected by fluorescence confocal microscopy using 2′,7′-dichlorofluorescin diacetate (DCFH-DA) as a marker. As shown in Figure 3, upon UV irradiation, the level of green fluorescence in untreated HEKa cells was clearly higher for the cells treated with the NPs, confirming the protective action provided by their formation of an artificial NP perinuclear cap, which could be easily observed by TEM.

In the case of X-ray irradiation, ROS production can be more severe than in the case of UV exposure. In order to better protect cells from this kind of irradiation, Gianneschi’s group functionalized artificial melanin NPs with a nitroxide radical [102]. One very peculiar feature of melanin biopolymers is their persistent electron paramagnetic resonance (EPR) signal due to the presence of intrinsic semiquinone-like doublet-state radicals, which originate from an equilibrium between quinone and hydroquinone that yields semiquinone. It has been proposed that these radicals are correlated with catalytic superoxide decomposition, thus justifying the use of melanin in nature as a nanoscale radiation protector. For example, melanized fungi were found to have colonized the walls and the cooling pool water at the Chernobyl nuclear reactor site. The intrinsically low radical content in natural and synthetic melanin can be increased by metal complexation. Nevertheless, since heavy metals generally raise concerns in biomedical applications, the use of a metal-free methodologies to manipulate the radical content of melanin was investigated. In particular, a melanin-mimic NP precursor was obtained via the amide coupling of 3-(3,4-dihydroxyphenyl)-l-alanine (l-DOPA) with 4-amino-TEMPO, and was polymerized upon oxidation with atmospheric oxygen in an alkaline (NH_4_OH) solution. The ROS-scavenging properties of cells treated with radical-containing NPs and exposed to X-ray irradiation were investigated by loading Primary Normal Human Epidermal Keratinocytes cells (NHEK) with the ROS-responsive fluorogenic probe, Dichlorodihydrofluorescein diacetate (DCFDA), prior to X-ray irradiation. As expected, Confocal Laser Scanning Microscopy (CLSM) revealed an intense green fluorescence after the X-ray irradiation of unprotected cells. On the contrary, for the radical melanin or PDA-treated cells, no fluorescence from the ROS probe was observed, indicating that these NPs are efficient ROS scavengers in NHEK cells.

The same research group also developed artificial allomelanin NP and demonstrated their radical scavenging activity [103]. Allomelanin refers to a group of melanins that consist of nitrogen-free precursors, such as catechol and 1,8-dihydroxynaphthalene (1,8-DHN); this was used as a precursor for the synthesis of artificial allomelanin NPs that were produced via the oxidative oligomerization of 1,8-DHN in aqueous solution at room temperature using NaIO_4_ and KMnO_4_ as oxidative agents. The AOX properties of allomelanin NPs were investigated using the DPPH assay. The scavenging activity was determined by monitoring the decrease in absorbance at 516 nm, indicative of the free radical DPPH. DPPH is reduced through an electron transfer from the AOX material. Allomelanin NPs show a much higher radical trapping activity than that of PDA NPs, with similar activity to ascorbic acid. For cellular testing, NHEK cells were treated for 3 days with either the vehicle or 0.02 mg/mL of allomelanin NPs or PDA NPs. They were subsequently incubated with 5/6-chloromethyl-2′,7′-dichlorodihydrofluorescein diacetate (CM-H2DCFDA), a pro-fluorescent dye that is ROS-responsive, and then directly exposed to 365 nm UV light for 2 min. As shown in Figure 4, the control/untreated cells showed a higher fluorescence signal than the allomelanin- or the PDA-treated cells, indicating that allomelanin NPs serve as an effective antioxidant inside the cells. In addition, it was qualitatively observed that cells treated with allomelanin NPs quenched ROS more efficiently than those with PDA NPs.

## 4. AOX Therapy Based on Melanin-like NP

The correlation between oxidative stress, aging, and diseases has been recently strongly supported and discussed [104]. Starting from this observation, AOXs have been proposed as possible therapeutic agents for treating severe pathologies. In this section, the most recent and interesting examples of the use of melanin-like NPs for AOX therapy are discussed.

Ai et al. reported the synthesis and characterization of the multicomponent system manganese dioxide encapsulated selenium–melanin (Se@Me@MnO_2_) nanozyme (where melanin is PDA) with high efficiency against intracellular antioxidation and anti-inflammation [105]. In vitro experiments showed that this Se@Me@MnO_2_ nanozyme exhibits multiple enzyme-like activities that are able to scavenge ROS. The mechanism that the researches illustrated was explained based on the fact that the Se core possesses catalytic activity similar to glutathione peroxidase, while the PDA and the MnO_2_ possess both the superoxide dismutase and the catalase-like activities. For the preparation of the multilayer system, the PDA section was obtained via the oxidation of dopamine with KMnO_4_. The HO^•^ scavenging efficiency of the nanozyme was measured by investigating its effect on the inhibition of the formation of fluorescent 2-hydroxyterephthalic from non-fluorescent terephthalic acid in the presence of H_2_O_2_. Additional studies were based on EPR upon the addition of H_2_O_2_: 5,5-dimethyl-1-pyrrolineN-oxide (DMPO), which was applied to capture HO^•^. After exposure to UV light for 3 min, the adducts DMPO-HO^•^ were collected and detected by the EPR. In order to test the intracellular ROS scavenging capability of the nanocomposite, oxidative stress was induced in hepatoblastoma (HepG2) cells. In the presence of the nanoenzyme, the cellular death produced by the oxidative stress condition was considerably reduced. In vivo experiments were performed to examine the potential application of the nanoenzyme in blocking ROS-triggered inflammation. Ear inflammation was induced in Kunming mice via the injection of phorbol 12-myristate 13-acetate. Inflammation produced in this way is easily detectable since 6 h after injection, the mouse ear turns to a black color. The effect of the injection of different nanomaterials (Se, Me, Se@Me MnO_2_, and Se@Me@MnO_2_ nanozyme) on the inflamed site was investigated. The best result was obtained with the Se@Me@MnO_2_ nanozyme, which caused a clear decrease in the dark color associated with the inflammation. A confirmation of the oxidative stress reduction resulted from the use of the probe DCFH-DA, which becomes fluorescent upon reaction with ROS. The green fluorescence of the inflamed mouse ear was very strong when detected by fluorescence microscopy, but became much weaker in the case of the Se@Me@MnO_2_ nanozyme-treated mouse groups.

Zhao et al. developed a molecular precursor suitable for in-situ antioxidative melanin formation and for acute liver injury therapy and imaging [106]. The precursor molecule was a PEGylated phenylboronic-acid-protected L-DOPA (PAD, shown in Figure 5) that, after deprotection in the presence of ONOO^−^, was demonstrated to self-assemble into well-defined NPs (PAD NP) following a biosynthetic pathway similar to the one of melanin. PAD NPs were efficient in the prevention and treatment of acute liver injury/failure, and their in situ formation was exploited in order to visualize the damaged tissue non-invasively by photoacoustic imaging. The AOX activity of PAD was first tested at the cellular level in the Raw 264.7 (murine macrophages) cell line, using lipopolysaccharide (LPS) as an activator of macrophages to induce the production of oxidative species, such as H_2_O_2_, ^•^OH, ONOO^−^, and ^•^NO. In particular, intracellular ^•^NO was measured using the Griess reagent assay kit, proving that the PAD NPs effectively reduced the excess ^•^NO in activated Raw 264.7 cells in a concentration-dependent manner. More generally, intracellular ROS were detected with a DCFH-DA reagent assay kit using flow cytometry. Experiments demonstrated that ROS was significantly increased in Raw 264.7 cells after using an LPS trigger, but also that PAD NPs effectively inhibited the overproduction of ROS in LPS-treated macrophages. In vivo testing was performed on a group of drug-induced liver-injured (DILI) mice. Mice were injected by acetaminophen to cause acute liver injury and then were treated with PAD. The formation of PAD NPs in the liver of the acetaminophen-injected mice could be detected by photoacoustic imaging, while no signal was detected in healthy mice. Moreover, two common clinical indicators of liver function, alanine aminotransferase and aspartate aminotransferase, were analyzed to prove that acetaminophen treatment caused acute hepatic dysfunction, which is a state that causes a relevant increase in both these liver enzymes. Interestingly, treatment with PAD displayed obvious efficacy in reducing enzyme levels in a dose-dependent manner in DILI mice, while treatment with the conventional molecular antioxidant NAC (N-Acetyl-Cysteine) produced no statistically significant decrease.

Sun et al. demonstrated the AOX properties of a melanin-based natural NPs for theragnostic application in Acute Kidney Injury (AKI), a pathology associated with oxidative stress that is able to produce high mortality in clinics [107]. As schematized in Figure 6, these NPs were prepared from melanin granular powder, Mn^2+^ and polyvinylpyrrolidone (PVP) dissolved in dimethyl sulfoxide (DMSO). Thanks to the poor water solubility and strong chelating capabilities of melanin and PVP, a self-assembly process was initiated to form water-soluble NPs by adding the above mixture to deionized water at a 1:10 (*v*/*v*) ratio. To improve the physiological stability, thiol-terminated polyethylene glycol (HS-PEG) was incorporated via a Michael addition reaction.

The ability of the NPs to act as ROS scavengers in vitro was tested for two physiologically relevant ROS: the O_2_^•−^ and ^•^OH. Experiments showed that the NP were highly sensitive to O_2_^•−^, showing successful scavenging of 79.4 ± 4.7% of O_2_^•−^ at a melanin concentration 25 μg mL^−1^ and the scavenging of 68.4 ± 2.5% of ^•^OH at a melanin concentration of 100 μg mL^−1^. Furthermore, the NP-mediated free radical scavenging ability was evaluated using the ABTS assay, demonstrating the elimination of 85.2 ± 2.3% of ABTS^•+^ at a melanin concentration of 100 μg mL^−1^. At the cellular level, the ROS-scavenging ability was investigated in human embryonic kidney 293 (HEK293), inducing oxidative stress via the administration of H_2_O_2_. Cells pretreated with the NPs at various concentrations exhibited significantly decreased levels of intracellular ROS in a dose dependent manner following H_2_O_2_ treatment. In vivo experiments were performed on mice by triggering injury related to oxidative stress in kidneys and renal dysfunction through water deprivation and the subsequent intramuscular injection of 50% glycerol. Subsequently, the effect of the intravenous injection of NPs on AKI was investigated using mice injected with PBS as controls. Indeed, severe body weight loss was observed only in the PBS (but not in the NP)-treated AKI mice. The renal accumulation of the NP was demonstrated both by Magnetic resonance (MR) imaging, exploiting the Mn reach particles as contrast agents, and by Positron emission tomography (PET) imaging, after activating the NP with ^89^Zr. Additionally, clinical indices of kidney excretory function, and thus blood urea nitrogen (BUN) and serum creatinine levels, were analyzed to evaluate the therapeutic effect of the NPs. Results revealed that PBS treated AKI mice exhibited increased BUN and serum creatinine levels, typical hallmarks of renal failure. Importantly, NP treatment decreased the BUN and serum creatinine levels in AKI mice, and the renal function of AKI mice was able to be totally restored.

AOX therapy was also proposed for the treatment of SARS-CoV-2 infection [108]. In SARS-CoV-2 infection, immune cell infiltration creates an inflammatory and oxidative microenvironment, which can cause pneumonia, severe acute respiratory syndrome, kidney failure, and even death. SARS-CoV-2 virus mainly binds to the ACE2 (angiotensin converting enzyme 2) receptor, which is expressed on the alveolar epithelial cells through its surface spike protein (S protein) in order to enter the human host cell and complete the virus life cycle. Ma et al. developed a nano-bait based on the exosome-sheathed PDA (PDA@Exosome) NPs generated via the exocytosis of the PDA NPs from H293T cells. The resulting PDA@Exosome can compete with ACE2-expressing epithelial cells for S protein binding and can significantly attenuate the level of inflammatory cytokines by mediating oxidative stress, a major cause of organ injury. As the ACE2 protein receptor cluster is the basis for the nano-bait’s competitive binding ability, 293T cells that highly express ACE2 receptors were constructed. The AOX activity of the PDA core in the nano-bait was investigated. The scavenging efficiency of PDA NPs toward HO^•^ followed a concentration-dependent manner and could be enhanced with increasing the concentration of PDA. HO^•^ scavenging efficiency was achieved by measuring the presence of fluorescent 2-hydroxyterephthalic acid resulting from the capture of HO^•^ by terephthalic acid. O_2_^•−^ scavenging efficiency was achieved by measuring the inhibition ratio of photoreduction, upon UV irradiation, of nitro blue tetrazolium (NBT) in a solution containing riboflavin, methionine and NBT. More than 90% of O_2_^•−^ could be removed from the solution in the presence of PDA NPs with a concentration of 0.2 mg mL^−1^.

Ischemic stroke is one of the leading causes of long-term disability and mortality. The infusion of Mesenchymalstem cells (MSCs) is a potential treatment for ischemic stroke, however transplanted MSCs show a low survival rate; this is related to many factors, including intense inflammation and oxidative stress [109]. NPs were prepared by the simple atmospheric oxidation of dopamine in an alkaline (NaOH, pH = 8.5) solution. The free radical-scavenging capacity of NPs was detected by the DPPH scavenging method.

Primary cortical neurons were cultured and submitted for oxygen–glucose deprivation (OGD). Intracellular ROS levels were significantly increased in the OGD-vehicle-treated group compared with the normal control group. To determine the antioxidative effect of NP–MSCs, an ROS/superoxide detection assay was used to detect the intracellular ROS and superoxide generation in neurons after OGD exposure. Treatment with NP–MSC, MSC, or NP also reduced the superoxide levels in OGD-damaged neurons by 52.26%.

Rheumatoid arthritis (RA) is an autoimmune disease characterized by synovial dysplasia and chronic joint inflammation, and it can be treated by combined antioxidative therapy and chemotherapy [110]. For this, the efficient delivery of drugs and AOXs to the RA synovial joint is fundamental. Wu et al. proposed a programmable polymeric microneedle (MN) platform for the transdermal delivery of methotrexate (MTX) and PDA@MnO_2_ as a ROS-scavenging agent (Figure 7).

The polymeric platform was able to degrade into skin tissues releasing loaded MTX and PDA@MnO_2_. The PDA@MnO_2_ was demonstrated both to be an efficient MR imaging contrast agent and to work as a robust AOX in order to scavenge ROS and, in combination with the action of MTX, to decrease RA inflammation. The antioxidative performance of PDA@MnO_2_ was investigated by testing its ability to scavenge HO^•^ and O_2_^•−^. The HO^•^ scavenging efficiency was measured by using a 2-hydroxyterephthalic acid probe. The O_2_^•−^ scavenging efficiency was evaluated using the inhibition ratio of the photoreduction of NBT. After UV irradiation in the presence of riboflavin, methionine and NBT and in the absence of NPs, a strong fluorescence signal was produced, indicating a high degree of superoxide. On the contrary, in the presence of the NPs, the fluorescence signal decreased dramatically, demonstrating the removal capacity of O_2_^•−^. The AOX activity was also investigated in RAW264.7 by using lipopolysaccharides (LPS), which were then added into the cell culture medium to generate the intracellular ROS that were detected using DCFH-DA (2,7-dichlorofluorescein diacetate) as a fluorogenic probe. As expected, LPS-treated cells showed a strong fluorescence signal, because of the oxidation of DCFH to fluorescent 2,7-dichlorofluorescein (DCF) by ROS. The fluorescence almost completely disappeared in the cells with PDA@MnO_2_, proving that these NPs suppress ROS efficiently.

Osteoarthritis (OA) is characterized by progressive cartilage degradation, subchondral bone remodeling, and osteophyte formation. Wu et al. showed that AOX PDA-PEG NPs can be used to alleviate early osteoarthritis [111]. PDA-PEG NPs were synthesized from dopamine hydrochloride in deionized water in the presence of NaOH. After 2 h, the PDA NPs were collected by centrifugation, washed, and functionalized with mPEG-NH_2_, and purified by further centrifugation. The AOX activities of PDA-PEG NPs in water and organic solvent were evaluated by ABTS and DPPH assays, respectively. The antioxidant activities of PDA-PEG NPs were calculated referring to Trolox, which was used as a standard. Electron Spin Resonance Spectroscopy (ESR) was also utilized to assess the ability of PDA-PEG NPs to scavenge ROS in an acellular environment. Moreover, ROS in bone marrow-derived monocytes (BMMs) were measured by a DCFH-DA probe. First, the cytotoxic effects of H_2_O_2_ on BMMs were investigated, showing that concentrations of H_2_O_2_ higher than 200 µM inhibited cell proliferation. At lower concentrations, the exogenous administration of H_2_O_2_ was demonstrated in order to promote osteoclastogenesis. Interestingly, experiments demonstrated that PDA-PEG NPs could attenuate H_2_O_2_-induced osteoclastogenesis.

The AOX properties of melanin NPs were exploited also for the treatment of Parkinson’s disease [112]. In this pathology, the overproduction of reactive oxygen species (ROS) and cumulative oxidative stress induce the degeneration of neuromelanin-containing dopaminergic neurons in substantia nigra pars compacta (SNpc) patients. PDA is known to mimic the activities of superoxide dismutase and catalase in removing ROS. 

GPX is important for maintaining ROS metabolic homeostasis, but only a few GPX-like nanozymes have been studied for in vivo therapy. Since selenocysteine (SeCys) is essential for the antioxidant activity of GPX, Wang et al. prepared a new nanocomposite with GPX-like activity. These presented the same CAT and SOD enzymatic activities as PDA, but showed better free radical scavenging efficiency and additional GPX enzymatic activity that could increase intracellular GPX levels effectively. PDA and PDASeCys NPs were synthesized in Tris buffer at pH 8.5, as schematized in Figure 8. Compared to PDA synthesized under the same conditions, the size of PDASeCys NPs was smaller, and the particle size decreased gradually with the increase in selenium content. The free radical scavenging ability of the material can reflect its antioxidant capacity. Therefore, DPPH^•^ and ABTS^+•^ were selected to evaluate the free radical scavenging rate of the NPs.

To test the activity at the cellular level, a subclone of the SK-N−SH neuroblastoma cell line (SHSY5Y) was used, since these kinds of cells have the properties of dopaminergic neurons and can express tyrosine hydroxylase (TH), which is widely used in the establishment of pathological models of (programmed death) PD cells. In particular, the cells were treated with a 1-methyl-4-phenyl-pyridine ion (MPP^+^), used as a neurotoxin to establish the cell model of PD. First, SHSY5Y cells were co-incubated with MPP^+^ at increasing concentrations for 24 h, and MTT ((3-(4,5-Dimethylthiazol-2-yl)-2,5-diphenyltetrazolium bromide) was used to detect cytotoxicity; as such, cell activity gradually decreased with the increase in MPP^+^ concentration. When the concentration of MPP^+^ was 2 and 3 mM, cell activities decreased to 50 and 40%, respectively. These two concentrations were chosen as reference values in order to investigate the effect of the NPs. At the highest concentration of MPP^+^, the SeCys-containing NPs allowed a recovery of up to 90% of the cellular functionality, while in the case of bare PDA NPs, the recovery was only up to 60%. The difference in the effect of the SeCys and Se-free NPs was even more evident when treating the cell with both MPP^+^ (3 mM) and a GPX inhibitor (RSL3) (4 μM). In this case, the Se-containing NP could recover the GPX activity to the normal level, while the Se-free NPs showed no effect. The production of oxidative stress induced by MPP^+^ was demonstrated by flow cytometry using a ROS fluorescent probe called DCFH-DA. The results showed that MPP^+^-treated SH-SY5Y cells generated 2.9-fold more ROS than the control. After the PDA and SeCys-PDA NP treatments, the level of ROS was reduced to 1.7-fold and 1.5-fold, respectively.

AOXs are known to improve the wound healing process and are investigated as a promising therapeutic strategy to treat chronic wounds [113,114,115,116,117,118,119,120,121,122]. Different kinds of PDA-based hydrogel formulations have been proposed for this application. In particular, using polyethylene imine (PEI) as a base, O’Connor et al. synthesized PDA in the form of a PEI-PDA copolymer [123]. Dextran was then functionalized with epoxy groups and crosslinked to the PEI-PDA copolymer to produce dark gels. The AOX activity of PDA-containing hydrogel was confirmed to be dependent on the amount of dopamine used in hydrogel synthesis. An antioxidant and antibacterial scaffold for rapid angiogenesis and diabetic wound repair was proposed by Tu et al., who fabricated a hydrogel through the dynamic crosslinking between polypeptide and polydopamine and graphene oxide, as schematized in Figure 9 [124]. 

Wang et al. developed a bioactive hydrogel made by collagen and hyaluronic acid by fabricating different PDA-modified lyophilized collagen hyaluronic acid copolymers [125]. Ren et al. synthesized ε-polylysine-grafted nanocellulose and PDA NPs, and embedded them in genipin-cross-linked gelatin to prepare a hydrogel [126].

Fu et al. incorporated PDA NPs into oxidized dextran/chitosan hybrid hydrogels [15]. The resulting hydrogels were demonstrated to have excellent AOX properties, to protect cells against external oxidative stress and to protect the wound against infections. Ge et al. developed a thermo-sensitive hydrogel that undergoes a sol–gel transition at body temperature and behaves as a ROS scavenger [16]. This hydrogel is based on a PDA-modified poly (ε-caprolactone-co-glycolide)-b-poly(ethylene glycol)-b-poly(ε-caprolactone-co-glycolide) triblock copolymer. The ROS-scavenging ability was demonstrated by DPPH and ABTS assays and intracellular ROS downregulation in RAW 264.7 cells. Li et al. exploited PDA coating to enhance the antioxidant abilities of MXene when preparing an injectable hydrogel based on hyaluronic acid-grafted dopamine and PDA-coated Ti3C2 MXene nanosheets. Wang et al. combined poly-sulfobetaine methacrylate, which is a representative zwitterionic polymer that is used for the preparation of hydrogels, by reinforcing it with 2,2,6,6-tetramethylpiperidinyl-1-oxyl-oxidized nano-fibrillated cellulose [127]. The AOX activity and good biocompatibility was achieved by introducing Zn ion-loaded PDA-coated nano-fibrillated cellulose. In another example, gelatin-grafted dopamine and PDA-coated carbon nanotubes were used to engineer antibacterial, adhesive, AOX and conductive composite hydrogels through the oxidative coupling of catechol groups using a horseradish peroxidase catalytic system [17].

## 5. Melanin and Cancer

The applications of natural melanin and melanin-like NPs in cancer therapy include chemotherapy, phototherapy, immunotherapy, and gene therapy [128,129,130,131]. Nevertheless, the role of melanin in cancer is still widely debated. This is due to the dual role that melanin has been reported to play in the initiation and progression of melanoma [132,133]. Melanin, naturally present in human skin, is in fact known to behave as a photo-protecting agent, a role that involves both the ability to absorb light, especially the UV that can create DNA damage via the photochemical generation of CPDs or other photoproducts [99], but also the capacity to quench radicals and reactive species generated by the light irradiation of living tissues. Indeed, recent studies have revealed that the irradiation of melanin can trigger DNA mutations. According to Premi et al., the mechanism involved is “chemiexcitation” [53,134], a process that starts with the formation of the strong oxidant peroxynitrite (ONOO−) from the UV radiation induced in the radicals NO and O_2_^-^. Hence, peroxynitrite oxidizes melanin that binds O_2_ with the formation of a dioxetane, which is a high-energy 4-membered ring containing two adjacent oxygen atoms that can thermally cleave to create two carbonyls. Very importantly, because of the high amount of energy available, one of the two carbonyl residuals produced is in a triplet electronically excited state. The energy of this long lived excite can be donated to DNA via a Dexter exchange mechanism, thus producing CPDs. These mutations, induced by chemiexcitation, can be produced a long time after UV exposure; thus, when actually in the “dark”, these are termed as “dark CPDs” (dCPDs). It is interesting to observe that according to the mechanism of formation, dCPDs can be prevented using species, and, in particular, AOXs, that are able to quench the triplet-excited state of carbonyl. Additional complications related to melanin were found in melanoma therapeutic treatment since melanin pigment can attenuate the efficacy of chemotherapy and radiotherapy [130,132].

PDA NPs have been used to induce ferroptotic cell death in cancer cells by exploiting their metal ions chelating ability. Considering this, PDA NPs, stabilized by PEG chains, were loaded with Fe^2+^ or Fe^3+^, reaching concentrations as high as 0.7–0.8 mg of iron per mg of nanomaterial, and then were used to induce ferroptosis in both tumoral cells and mice models [135].

## 6. Future Perspectives

The popularity of the use of melanin-like NPs in nanomedicine, and, in particular, for AOX therapy, results from the ease of preparing highly biocompatible artificial synthetic melanin-like NPs or shells through fast, effective and environmentally compatible procedures. Although the chemical and structural characterization of these synthetic species is often simpler than that of the natural ones, a complete understanding is still missing. This is mainly due to the complexity of the system and the presence of different chemical units that interact both via covalent and non-covalent bonds. For these reasons, investigating and comparing the AOX properties of different melanin-related NPs would be extremely important for the understanding of the following:

Amount of residual molecular precursor. Although the procedures for producing melanin-like materials involve a purification step that aims to remove the molecular precursor (e.g., dopamine in the case of PDA), it should be taken into account that these molecular precursors are usually very efficient AOX species. Hence, the presence even of a fraction of these precursors in the final product may lead to an over-estimation of the AOX properties of the NPs. Purification of the NPs should be performed very carefully, repeating several times the purification steps and checking the possible presence of unreacted precursors.

Composition of the NPs. The structure of the different kinds of melanin-like NPs is still strongly debated. For a better understanding of the AOX activity of different NPs, it would be very important to know, at least, the degree of oxidation of the different molecular units and, in particular, the faction of –OH units and carbonyl groups present in the aromatic part of the structure. Carbonyl groups in fact result from the oxidation of –OH, however they present a very different reactivity with respect to radicals, as is known for the simple quinone–hydroquinone pair. Without this information, it becomes difficult to understand the AOX behavior of the melanin NPs and the different activities they exert against different kinds of radicals.

Mechanism of the AOX activity of allomelanin. Most nanosized AOXs are based on the incorporation of PDA. Nevertheless, recent studies have suggested that allomelanin offers better AOX properties than PDA, being still highly biocompatible and obtainable via very versatile procedures. In principle, allomelanin could represent an advantageous substitute for PDA. Nevertheless, while the actual mechanism at the basis of the action of the AOX properties of PDA has been investigated, a detailed study on the interaction of allomelanin with different radicals that proves its superior AOX activity is still missing.

Interaction of melanin NPs with ROO radicals. Most studies about the AOX properties of melanin-like NPs are based on the quenching of O_2_^•−^, H_2_O_2_ or HO^•^. Meanwhile, on the contrary, there are very few examples of studies on the interaction with ROO^•^. In this context, it should be recalled that indeed ROO^•^ are involved in the processes of oxidizing organic matter using atmospheric oxygen, the process that most AOXs have been developed to prevent. Particular attention should be therefore paid to this topic.

Mechanism of quenching of H_2_O_2_. Although in several cases the ability of melanin-like NPs to quench a H_2_O_2_ has been reported, the mechanism of the process has not been discussed yet. In particular, it is not clear whether or not the process involves the intermediate formation of HO^•^ radicals and which modifications the NPs undergo because of the reaction.

Effect of metal chelation on the AOX properties. Melanin-like NPs are known to be able to chelate metal ions very efficiently. In some cases, they can even trap metal ions during synthesis when metal-based oxidants, such as permanganate, are used. The presence of a metal center is expected to affect the AOX behavior of melanin-based NPs. For example, the presence of metal ions can favor the Fenton reaction, which transforms H_2_O_2_ into HO^•^ radicals. Nevertheless, a systematic investigation of the effect of metal chelation on the AOX activity of melanin NPs is still missing. This should involve a very well characterized set of NPs with the same properties, but different amounts of chelated metal ions.

Considering the information just reported above, we believe that even if it is clear that there are incredible advantages to the incorporation of melanin as an AOX agent in NPs, some drawbacks are still present in this approach. In our opinion, these are mostly due to the lack of knowledge of some fundamental features of these unique materials. Considering the great effort that the scientific community is making in order to develop new, very effective and sophisticated platforms for nanomedicine, sometimes following a partially empirical approach, we would like to suggest that part of this community takes a step backwards and considers the basics of the properties, with the aim of better understanding these melanin-mimicking materials.

## 7. Conclusions

In this paper, we reviewed the most recent and relevant applications of melanin-mimicking NPs as AOX agents. The results discussed here prove the great potential of these materials to be used as AOX agents and their applicability in AOX therapy for the treatment of important and severe pathologies. Although this important conclusion needs to be taken into consideration, there is still substantial space to improve and deepen the understanding of the AOX properties of melanin. In fact, even if melanin-like materials have been known for a long time and have been widely exploited, their fundamental chemical properties, and even their actual structure, are still strongly debated. In addition to this,, due to their unique functional properties and their versatility, they have been incorporated into very effective nano-systems for nanomedicine, providing important characteristics, including AOX activity. The recent examples discussed in this review paper clearly demonstrate the huge potentialities of this approach, based on the exploitation of the AOX properties of melanin-like materials in nanomedicine.

## Figures and Tables

**Figure 1 antioxidants-12-00863-f001:**
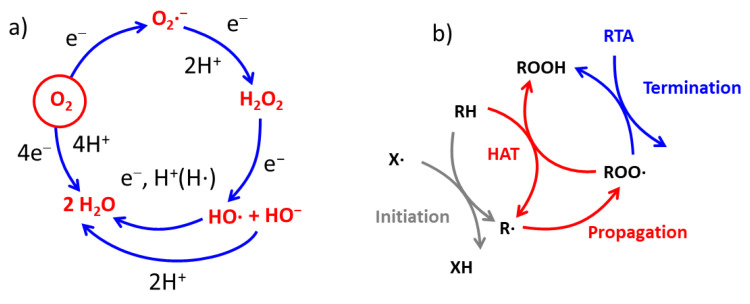
(**a**) Reduction processes involving the oxygen molecule, (**b**) propagation reaction leads to oxidation of organic molecules, RH, by oxygen.

**Figure 2 antioxidants-12-00863-f002:**
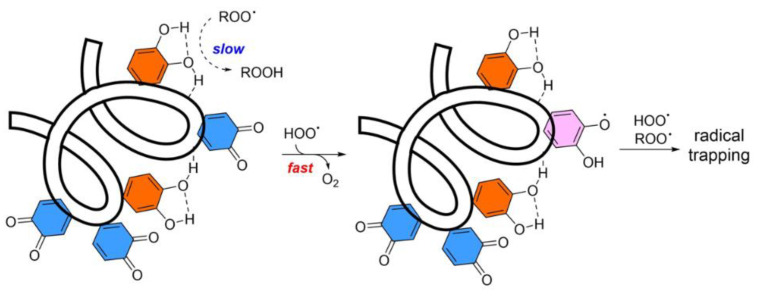
Catechol (orange) and quinone (blue) units in the polydopamine polymer are unreactive toward alkyl-peroxyl radicals (ROO·), but upon the reaction with HOO^•^, the quinones are converted to ortho-semiquinone radicals (pink) with an enhanced ability to trap both species, HOO^•^ or ROO^•^. Reprinted with permission from ref. [83].

**Figure 3 antioxidants-12-00863-f003:**
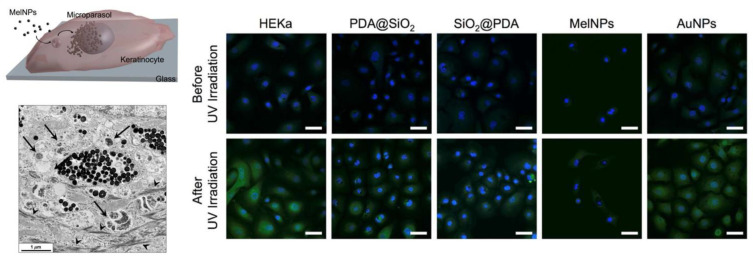
MelNPs uptake scheme and TEM image of HEKa cells with MelNPs after 3 days incubation: black arrows indicate melanosomes and black arrowheads indicate keratin fibers. Confocal imaging of ROS detection in HEKa cells with MelNPs, SiO_2_@PDA core–shell NP, PDA@SiO_2_ core–shell NP, and AuNPs after incubation for 3 days. Data are shown before and after 5 min of UV irradiation of these cells. The nuclei were stained with NucBlue (blue); ROS generated in HEKa cells were detected with DCFH-DA (green). Scale bars are 50 μm. Reprinted with permission from ref. [101].

**Figure 4 antioxidants-12-00863-f004:**
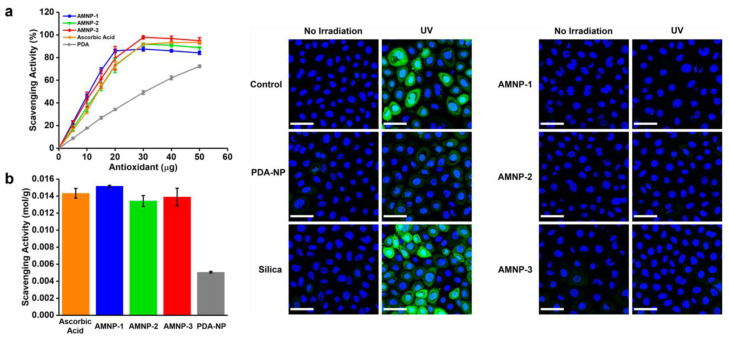
Radical scavenging by AMNPs compared to that of PDA-NPs and ascorbic acid. (**a**) DPPH radical scavenging activity of antioxidants; (**b**) calculated amount of quenched DPPH per gram of antioxidant. Oxidative stress was assessed via the ROS-activated CM-H2DCFDA dye (green). NHEK cells were incubated for 3 days with 0.02 mg/mL of AMNP-1, -2, and -3, PDA-NPs, silica NP, or the vehicle (water), treated with the dye, subjected to UV irradiation, and imaged live. Nuclei were stained with Hoechst (blue). Reprinted with permission from ref. [103].

**Figure 5 antioxidants-12-00863-f005:**
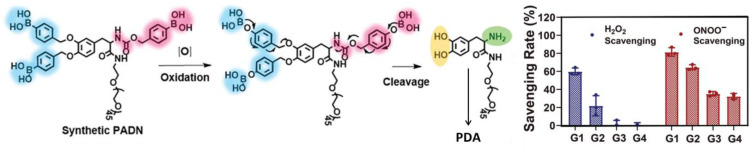
The designed PADN acts as a tyrosine mimic to scavenge reactive oxygen species (ROS) by oxidative polymerization similar to natural tyrosine. H_2_O_2_ and ONOO^−^-scavenging effect of PADN, PEG-Dopa, ONOO^−^-oxidized PADN and Mel-NPs, respectively (G1: PADN; G2: PEG-Dopa; G3: ONOO^−^-oxidized PADN; G4: Mel-NPs). Reprinted with permission from ref. [106].

**Figure 6 antioxidants-12-00863-f006:**
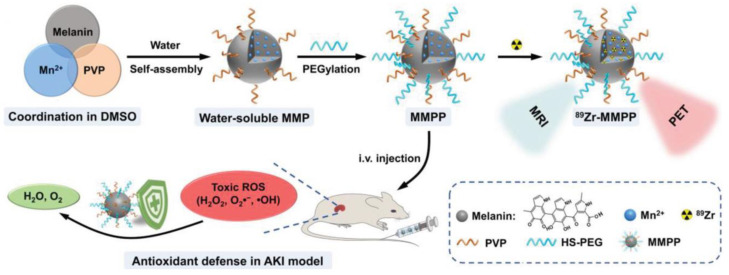
Synthesis and characterization of MMPP NP. Schematic illustration of the MMPP NP synthesis process and their activity as a naturally antioxidative platform for PET/MR bimodal imaging-guided AKI therapy. Reprinted with permission from ref. [107].

**Figure 7 antioxidants-12-00863-f007:**
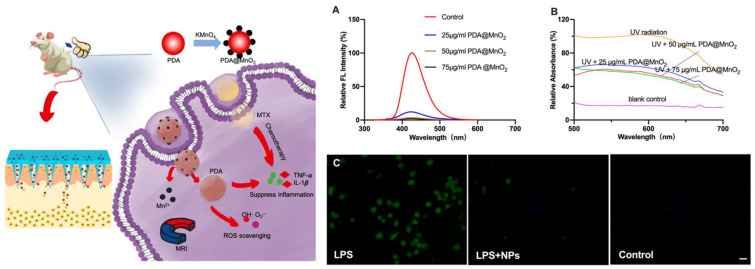
Schematic illustration of the synthesis of a high-performance nanozyme of PDA@MnO_2_ and the fabricated microneedles successfully delivered drugs with nanocomposites, as shown by results in paw swelling, MRI Image, and the level of cytokine. (**A**) scavenging efficiencies of OH^•^ and (**B**) O_2_^•−^ with PDA@MnO_2_. (**C**) ROS levels in the LPS-treated RAW264.7 cells with or without PDA@MnO_2_ treatment (control is without LPS and PDA@MnO_2_) detected by fluorescence images. Reprinted with permission from ref. [110].

**Figure 8 antioxidants-12-00863-f008:**
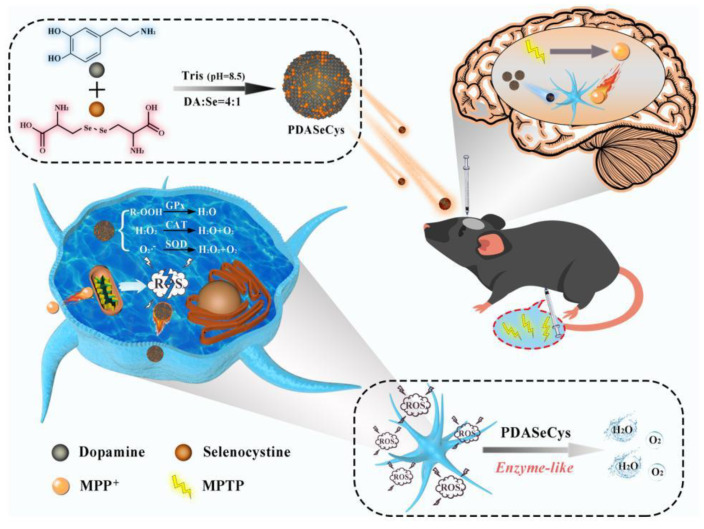
Synthesis process of PDASeCys NPs and their treatment for Parkinson’s Disease. Reprinted with permission from ref. [112].

**Figure 9 antioxidants-12-00863-f009:**
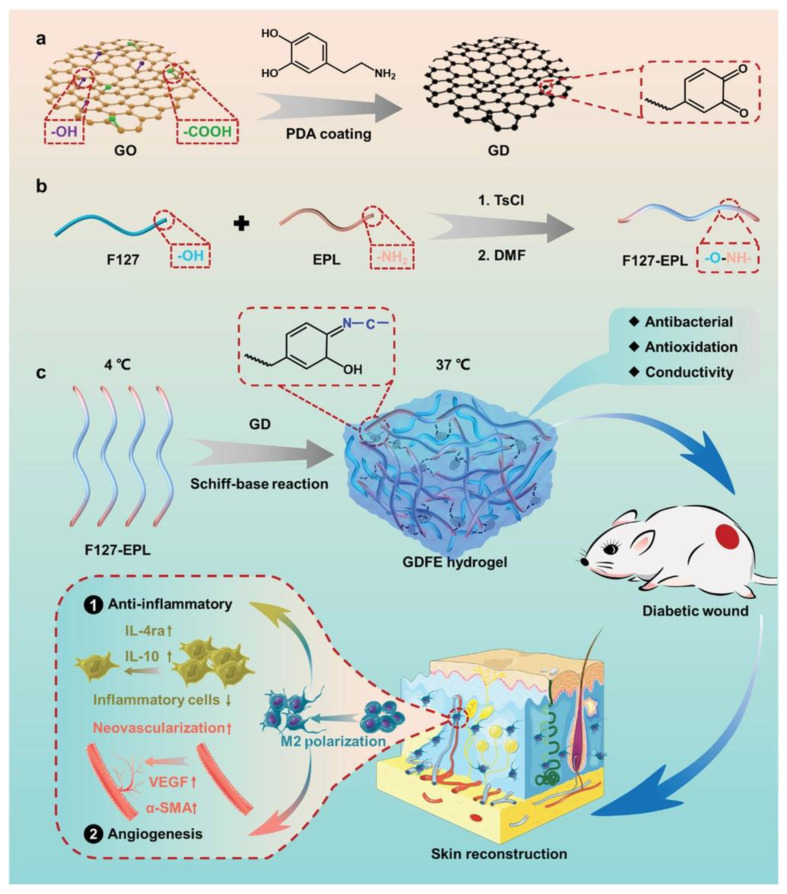
Schematic illustration of the synthesis of GDFE hydrogel with multifunctional properties for potential application in diabetic wound healing and skin reconstruction. (**a**) Schematic diagram of synthesis of GD. (**b**) Schematic diagram of synthesis of FE hydrogel. (**c**) The preparation, multifunctional properties and application of GDFE hydrogel. Reprinted with permission from ref. [124].

**Table 1 antioxidants-12-00863-t001:** Classification of antioxidant systems and main representative molecules, enzymes, or nanomaterials.

Class.	Subclass	Examples	Reference
Preventive	Iron chelation	Deferoxamine, ciclopiroxamine	[28]
	Peroxide decomposition	CAT, GPX, CAT-and GPX-like nanozymes	[27,29,30]
	Superoxide dismutation	SOD, SOD mimics, SOD-like nanozymes	[31,32]
Radical-trapping	Phenols	Tocopherols, polyphenols	[27,33]
	Quinones–hydroquinones	Ubiquinol	[27]
	Aromatic amines	Ferrostatin-1, phenoxazine	[28,34]
	Others	Ascorbate, nitroxides,	[35]

## Data Availability

Data sharing not applicable.

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
