# Peer review of "Recent Applications of Melanin-like Nanoparticles as Antioxidant Agents"

_antioxidants, 2023, doi:10.3390/antiox12040863_

Round 1

Reviewer 1 Report

In the work “Recent applications of melanin-like nanoparticles as antioxidant agents” the authors reviewed the potential of melanin NPs as antioxidant agents for different applications. The review is well-written and easy to follow. Some typos can be found (for example Page 2, line 57 – “the” instead of “thwe”). Also, some minor questions could be addressed:

-        Page 2, lines 82-83 – please include examples of the antioxidant regulation system (for example some enzymes).

-Pages  3 and 4 – authors could include a Table resuming the types of preventive and radical-trapping antioxidants.

-        Page 4, lines 172-173 – please give more information regarding each type of melanin.

-        Page 5 – Caption Figure 2 – identify the catechol and quinone units by the colors.

-        Increase the quality of Figure 8.

Author Response

Reviewer 1

In the work “Recent applications of melanin-like nanoparticles as antioxidant agents” the authors reviewed the potential of melanin NPs as antioxidant agents for different applications. The review is well-written and easy to follow. Some typos can be found (for example Page 2, line 57 – “the” instead of “thwe”).

Our reply>> We thank the reviewer for the positive comment and we corrected typos, in particular the on at page 2, line 57.

Also, some minor questions could be addressed:

Page 2, lines 82-83 – please include examples of the antioxidant regulation system (for example some enzymes).

Our reply>> In agreement with the comment of the reviewer in the new version of the manuscript we remand to section 2.1 for a detailed description of the antioxidant regulation system, also mentioning the new table the same reviewer suggested us to introduce in the next comment-

Pages  3 and 4 – authors could include a Table resuming the types of preventive and radical-trapping antioxidants.

Our reply>> In agreement with the comment of the reviewer in the revised version of the manuscript we included a totally new table (table 1) were different kinds and examples of antioxidants are summarized.

Page 4, lines 172-173 – please give more information regarding each type of melanin.

Our reply>> In agreement with the comment of the reviewer we added more information about the different kinds of melanin, redirecting to a specific paper for details.

Page 5 – Caption Figure 2 – identify the catechol and quinone units by the colors.

Our reply>> In agreement with the comment of the reviewer we identified catechol and quinone by color in the figure caption.

Increase the quality of Figure 8.

Our reply>> In agreement with the comment of the reviewer we increased the quality of figure 8.

Reviewer 2 Report

Author should confirm whether the images used in the article are original image or the permission is taken to reproduce the images in this article.

Language: At a few instances, the authors failed to comply with the international language standards, most of them are inadvertent.  The same should be improved upon.

There is a lack of spaces between words, which significantly encumber the perceiving of the text. Please, eliminate this drawback!.

*Abbreviations that are unavoidable must be defined at their first use in the manuscript. Ensure consistency of abbreviations throughout the article.

*Correct the italic style of ‘in vitro and ex vivo’ in all the text.

The figure legends and Tables caption need improvement. All legends and captions should have enough description for a reader to understand the figure/table without having to refer back to the main text of the manuscript.

Authors should at the minimum proof read the entire manuscript for typographical errors and fix all grammatical errors.

Author Response

Author should confirm whether the images used in the article are original image or the permission is taken to reproduce the images in this article.

Our reply>> We thanks the reviewer for the comment. Figure 1 is totally original while the other figures were reproduced with permission. In order to clarify this point we added to the caption Reprinted with permission from ref. xx

Language: At a few instances, the authors failed to comply with the international language standards, most of them are inadvertent. The same should be improved upon.

Our reply>> According to the suggestion of the reviewer we improved the quality of the language.

There is a lack of spaces between words, which significantly encumber the perceiving of the text. Please, eliminate this drawback!.

Our reply>> According to the suggestion of the reviewer we added spaces were needed

*Abbreviations that are unavoidable must be defined at their first use in the manuscript. Ensure consistency of abbreviations throughout the article.

Our reply>> According to the suggestion of the reviewer we checked the consistency of the abbreviations.

*Correct the italic style of ‘in vitro and ex vivo’ in all the text.

Our reply>> According to the suggestion of the reviewer we corrected the italic style of ‘in vitro and ex vivo’ in all the text.

The figure legends and Tables caption need improvement. All legends and captions should have enough description for a reader to understand the figure/table without having to refer back to the main text of the manuscript.

Our reply>> According to the suggestion of the reviewer we modified the legends and caption to make them understandable without referring back to the main text of the manuscript.

Authors should at the minimum proof read the entire manuscript for typographical errors and fix all grammatical errors.

Our reply>> According to the suggestion of the reviewer we proof read the manuscript and fixed the errors

Reviewer 3 Report

Dear Authors,

I write you in regard to the review manuscript entitled Recent applications of melanin-like nanoparticles as antioxidant agents. 

- please, among lines 47-51, just as a suggestion to valorize the sentence, would it be possible to add the oxidative stress in skin injury?

- consider revising lines 56-58. 

- some content of the lines 158-167 could be at the abstract to valorize it.

Overall, the subject of the review has merit to be investigated. That being registered, the text must be deeply revised since it seemed, in some level, difficult to follow. Several paragraphs, concerning the text structure, were confusing. It was not clear what was the NP of the melanin-like compound. How such NP would be incorporated into a vehicle to be administered was not explored. 

Author Response

Dear Authors,

I write you in regard to the review manuscript entitled Recent applications of melanin-like nanoparticles as antioxidant agents.

- please, among lines 47-51, just as a suggestion to valorize the sentence, would it be possible to add the oxidative stress in skin injury?

Our reply>> According to the suggestion of the reviewer we mentioned skin injury as a disease in which oxidative stress plays an important role.

- consider revising lines 56-58.

Our reply>> According to the suggestion of the reviewer we revised lines 56-58.

- some content of the lines 158-167 could be at the abstract to valorize it.

Our reply>> Following the suggestion of the reviewer part of lines 158-167 were anticipated in the abstract in the revised version of the manuscript.

Overall, the subject of the review has merit to be investigated. That being registered, the text must be deeply revised since it seemed, in some level, difficult to follow. Several paragraphs, concerning the text structure, were confusing.

Our reply>> Following the suggestion of the reviewer we generally revise the text to make it clearee

It was not clear what was the NP of the melanin-like compound.

Our reply>> We apologize for this lack of clarity. Indeed natural melanin is mostly present in organism in the form of NP as for example in the sack of cuttlefish or other mollusks. We clarified this point in the revised version of the manuscript in page 5. Additionally artificial melanin, e.g. polydopamine, is in most cases formed as NP, this point is also clarified in page 5 in the new version of the manuscript also explaining that melanin-like materials are in general artificial compounds that mimics natural melanin.

How such NP would be incorporated into a vehicle to be administered was not explored.

Our reply>> We apologize this point was not clear enough in the previous version and we thanks the referee for the comment. Indeed melanin NP requires no vehiculation and they can be directly administered as suspensions in physiological medium. This point is clearly discussed in the new version of the manuscript in page 5.

Reviewer 4 Report

The work presented to me for review, entitled "Recent applications of melanin-like nanoparticles as antioxidant agents, is a review work. The authors try to explain

 mechanism behind the interaction of melanin-like nanostructures with different radicals and highly reactive species. in my opinion, the work is interesting and presents a typical layout of a review paper. Abstract and Introduction is quite clear and generally introduces the reader to further considerations, however, in the Introduction I miss entering the specific purpose of the work which is the leitmotif. –please add. Individual sections are described in an interesting and specific way with examples. Unfortunately, I miss the section, ,,study design’’or ,,Inclusion and exclusion criteria’’. In this type of article, it is necessary for the reader to know in what time limits the authors did screening-please add. Another element that is a bit too controversial for me is the copying of figures from cited articles, I think the authors should consider removing these figures, or if they want to introduce figures, they should be made by themselves.  Section 5 should be a little more expanded on cancer and melatonin with specific examples in an in vitro and in vivo model. In addition, I suggest the authors to separate the conclusions and perspectives sections into "conclusions’’ and ,,future perspectives" because in this form it is too long and it is difficult to capture the most important elements. Besides, in my opinion, the article is interesting and brings new information on melanin in the context of oxidation research. The work requires re-reading and correcting some typos and punctuation errors. In the title, please remove the dot at the end.

Author Response

The work presented to me for review, entitled "Recent applications of melanin-like nanoparticles as antioxidant agents, is a review work. The authors try to explain mechanism behind the interaction of melanin-like nanostructures with different radicals and highly reactive species. in my opinion, the work is interesting and presents a typical layout of a review paper. Abstract and Introduction is quite clear and generally introduces the reader to further considerations,

Our reply>> We thank the reviewer for the positive comment.

however, in the Introduction I miss entering the specific purpose of the work which is the leitmotif. –please add.

Our reply>> We are sorry this important point was not clear in the original version of the paper. Following the suggestion of the referee in the revised version we clearly stated in page 4 that “The objective of this review paper is to show that melanin, melanin-like and melanin containing NP can be efficiently use for AOX therapy in the case of several serious diseases.”

Individual sections are described in an interesting and specific way with examples. Unfortunately, I miss the section, ,,study design’’or ,,Inclusion and exclusion criteria’’. In this type of article, it is necessary for the reader to know in what time limits the authors did screening-please add.

Our reply>> Following the suggestion of the reviewer we added, in the revised version of the manuscript, in page 4 a section clarifying the “study design” and the “inclusion/exclusion” criteria we chose and the time limits we considered.

Another element that is a bit too controversial for me is the copying of figures from cited articles, I think the authors should consider removing these figures, or if they want to introduce figures, they should be made by themselves.

Our reply>> We agree with the reviewer that the use of figures from existing papers should be clearly stated and should be done only with permission. For this we added, when necessary “Reprinted with permission from ref. xx” to the figure captions in the revised version of the manuscript-

Section 5 should be a little more expanded on cancer and melatonin with specific examples in an in vitro and in vivo model. In addition

Our reply>> According to the suggestion of the reviewer we added new example of the use of melanin-like nanomaterials for treating cancer.

I suggest the authors to separate the conclusions and perspectives sections into "conclusions’’ and ,,future perspectives" because in this form it is too long and it is difficult to capture the most important elements.

Our reply>> According to the suggestion of the reviewer we split the section in two parts.

Besides, in my opinion, the article is interesting and brings new information on melanin in the context of oxidation research. The work requires re-reading and correcting some typos and punctuation errors. In the title, please remove the dot at the end.

Our reply>> According to the suggestion of the reviewer we revised the manuscript to eliminate typos and punctuation errors and we removed the dot at the end of the title.

Reviewer 5 Report

The manuscript entitled - Recent applications of melanin-like nanoparticles as antioxidant agents – is a very well designed review of the therapeutic potential of melanin nanoparticles.

The authors introduces the manuscript by a detailed description of the role of the so called Reactive Oxygen Species, Oxidative Stress behavior and its association with  severe pathologies. Furthermore, generation of radicals and regulation in living organisms are clearly explained.

In the following chapters, the anti-oxidative function and in more detail, the most relevant applications of melanin-NP are discussed on the basis of recently published scientific papers. 

Finally, the review includes a competent conclusion and also points out challenges and future perspectives.

Thus, this manuscript considered the most promising and relevant innovations in this field, is pleasant to read and provides an excellent overview for the community. The authors cited primarily papers of the last view years and the selected figures are well suited to reflect the text. In this regard, it should be confirmed that all figures are permitted by the corresponding authors.

Author Response

The manuscript entitled - Recent applications of melanin-like nanoparticles as antioxidant agents – is a very well designed review of the therapeutic potential of melanin nanoparticles.

The authors introduces the manuscript by a detailed description of the role of the so called Reactive Oxygen Species, Oxidative Stress behavior and its association with severe pathologies. Furthermore, generation of radicals and regulation in living organisms are clearly explained.

In the following chapters, the anti-oxidative function and in more detail, the most relevant applications of melanin-NP are discussed on the basis of recently published scientific papers.

Finally, the review includes a competent conclusion and also points out challenges and future perspectives.

Thus, this manuscript considered the most promising and relevant innovations in this field, is pleasant to read and provides an excellent overview for the community. The authors cited primarily papers of the last view years and the selected figures are well suited to reflect the text. In this regard, it should be confirmed that all figures are permitted by the corresponding authors.

Our reply>> We thank the reviewer for the very positive comments. Regarding the figures we uploaded all the permissions and we further add to the captions “Reprinted with permission from ref. xx” when necessary in the new version of the manuscript.

Round 2

Reviewer 3 Report

Dear Authors,

Thank you for addressing all questions.

Reviewer 4 Report

A accept authors' answers.